# Metabolome Changes in Cerebral Ischemia

**DOI:** 10.3390/cells9071630

**Published:** 2020-07-07

**Authors:** Tae Hwan Shin, Da Yeon Lee, Shaherin Basith, Balachandran Manavalan, Man Jeong Paik, Igor Rybinnik, M. Maral Mouradian, Jung Hwan Ahn, Gwang Lee

**Affiliations:** 1Department of Physiology, Ajou University School of Medicine, Suwon 16499, Korea; catholicon@ajou.ac.kr (T.H.S.); ekdus93@ajou.ac.kr (D.Y.L.); shaherinb@aumc.ac.kr (S.B.); bala@ajou.ac.kr (B.M.); 2College of Pharmacy, Sunchon National University, Suncheon 57922, Korea; paik815@scnu.ac.kr; 3Department of Neurology, Rutgers - Robert Wood Johnson Medical School, New Brunswick, NJ 08854, USA; igor.ryb@gmail.com (I.R.); mouradmm@rwjms.rutgers.edu (M.M.M.); 4Department of Emergency Medicine, Ajou University School of Medicine, Suwon 16499, Korea; 5Department of Molecular Science and Technology, Ajou University, Suwon 16499, Korea

**Keywords:** cerebral ischemia, metabolomics, metabolic network, middle cerebral artery occlusion, oxygen-glucose deprivation

## Abstract

Cerebral ischemia is caused by perturbations in blood flow to the brain that trigger sequential and complex metabolic and cellular pathologies. This leads to brain tissue damage, including neuronal cell death and cerebral infarction, manifesting clinically as ischemic stroke, which is the cause of considerable morbidity and mortality worldwide. To analyze the underlying biological mechanisms and identify potential biomarkers of ischemic stroke, various in vitro and in vivo experimental models have been established investigating different molecular aspects, such as genes, microRNAs, and proteins. Yet, the metabolic and cellular pathologies of ischemic brain injury remain not fully elucidated, and the relationships among various pathological mechanisms are difficult to establish due to the heterogeneity and complexity of the disease. Metabolome-based techniques can provide clues about the cellular pathologic status of a condition as metabolic disturbances can represent an endpoint in biological phenomena. A number of investigations have analyzed metabolic changes in samples from cerebral ischemia patients and from various in vivo and in vitro models. We previously analyzed levels of amino acids and organic acids, as well as polyamine distribution in an in vivo rat model, and identified relationships between metabolic changes and cellular functions through bioinformatics tools. This review focuses on the metabolic and cellular changes in cerebral ischemia that offer a deeper understanding of the pathology underlying ischemic strokes and contribute to the development of new diagnostic and therapeutic approaches.

## 1. Introduction

Cerebral ischemic disease, a common form of stroke, remains one of the leading causes of morbidity and mortality worldwide. Cerebral ischemia is the fifth leading cause of death and disability impacting one million Americans every year [1]. It is caused by blockage of a blood vessel due to a thrombus or embolus [2]. During cerebral ischemia, part of the brain lacks oxygen and nutrients, becoming severely damaged through induction of metabolic and cellular disturbances. Intravenous thrombolysis with tissue plasminogen activator (tPA) and catheter-based reperfusion therapies in appropriately selected patients are the main treatments to minimize tissue damage. Yet, many patients do not seek or have access to such treatments in time, leaving them with significant brain damage and disability. Therefore, understanding the pathological mechanisms and metabolic biomarkers of cerebral ischemia have been pursued using patient samples, animal models, and in vitro experiments [3,4,5,6,7,8,9,10,11,12,13,14,15].

In this review, we summarize the diagnosis of cerebral ischemia and the methodologies used to screen metabolic and cellular biomarkers. In addition, we introduce analytical platforms, including quantification platforms, statistical analyses, and bioinformatic tools. Finally, we summarize the metabolic changes and biomarkers in patients with cerebral ischemia, and in in vivo and in vitro models. 

## 2. Approaches to Diagnosis and Investigation of Cerebral Ischemia

Cerebral ischemia can be caused by hypoperfusion or cessation of cerebral perfusion due to (i) embolic or thrombotic occlusion of a cerebral artery inducing a localized stroke, (ii) cardiac arrest causing global cerebral hypoxia, or (iii) delayed cerebral ischemia in the setting of vasospasm following subarachnoid hemorrhage (SAH) [16,17,18,19,20]. Depending on the size, location, and duration of cerebral ischemia, irreversible damage of neuronal cells is established within 5–10 min after cessation of cerebral perfusion [4]. Disability occurs in 1 out of 100 patients for every 5-min delay in endovascular reperfusion treatment [21], with a 10.6% decrease in the probability of a favorable outcome due to a 30 min delay [22]. Therefore, timely, accurate diagnosis and decision-making regarding reperfusion is a critical factor in reducing the incidence and severity of disability caused by ischemic strokes [4,17,21,22,23,24,25]. 

In ischemic stroke, diagnosis and the decision to proceed with intravenous and catheter-based reperfusion therapies are made by considering the severity of clinical deficits with the National Institutes of Health Stroke Scale (NIHSS), imaging studies such as non-contrast computed tomography (CT) and CT-based Alberta Stroke Program Early CT Score (ASPECTS) to assess for early ischemic changes, CT angiography to confirm intracranial proximal large vessel occlusion, and multimodal imaging with CT or magnetic resonance imaging (MRI) perfusion studies to estimate potentially viable brain tissue where appropriate [25,26]. Despite having teams specialized in cerebrovascular disease, stroke cases can generate diagnostic difficulties, and the diagnosis of acute ischemic stroke may be delayed in up to 30% of patients especially with subtle clinical findings [3,27]. In the diagnosis of cerebral ischemia associated with SAH and cardiac arrest, there are critical hurdles such as a mismatch between angiographic vasospasm and regional hypoperfusion, limited clinical evaluation in the setting of impaired consciousness, and toxic and metabolic confounders [16,17]. Given all these difficulties, timely availability of bedside diagnostic and prognostic tests based on panels of biomarkers would be useful information when managing patients with cerebral ischemia. Even with an increasing knowledge of cerebral ischemia, advances in imaging techniques, a better understanding of risk factors, and improvements in the treatment guidelines and organization of medical teams, timely and accurate diagnosis of cerebral ischemia remains challenging [3,16,17,20,25,26,27,28,29,30]. Additionally, the sensitivity of non-contrast CT, CT angiography, and CT perfusion is highly variable [31]. Although MRI has higher sensitivity and specificity than CT for ischemia, limited availability, technical challenges, and cost makes MRI-based studies less practical in the acute setting. [32]. Therefore, an understanding of cerebral ischemia through more sophisticated metabolic and cellular studies, leading to the development of rapid and accurate analytical diagnostic biomarkers are needed.

In order to understand the pathological mechanisms of ischemic stroke, factors have been investigated in neurovascular unit (comprising elements of the blood–brain barrier, myocytes, pericytes, extracellular matrix, astrocytes, and neurons) [3,4]. Potential biomarkers of cerebral ischemia have been explored such as DNA [5], RNA [6], microRNAs (miRNAs) [7,8,9,10], proteins [11,12], and metabolites in patients’ plasma [13,15] or in brain tissue of rats with middle cerebral artery occlusion (MCAo) [14]. For example, elevated plasma DNA concentrations (>2500 kilogenome-equivalents/L) were detected in patients with hemorrhagic strokes compared with non-hemorrhagic strokes, with a 31% sensitivity and 83% specificity for discriminating the two types of stroke [5]. Plasma levels of miR-124-3p, miR-125b-5p, and miR-192-5p, which were found elevated using microRNA array analysis, correlated positively with infarct volume [7,8,9,10]. Protein biomarkers for endothelial damage, brain injury, inflammation, and coagulation/thrombosis were categorized and used to discriminate hemorrhagic stroke from ischemic stroke patients [12]. In the case of protein biomarkers in the rat MCAo model, different expression patterns of annexin A3 and synaptosomal-associated protein-25 were detected in brain tissue using 2-dimensional electrophoresis and matrix-assisted laser desorption/ionization-time of flight (MALDI-TOF) mass spectrometry (MS) [11].

From the perspective of therapeutic considerations, preclinical studies addressing the neuroprotective potential of manipulating certain metabolites associated with various pathways of cerebral ischemia include excitotoxic metabolites using a peptide inhibitor of c-Jun [33], anaerobic glycolysis-induced lactic acidosis using dichloroacetate [34] or induction of normoglycemia [35], and proinflammatory pathway mediators (lysophosphatidylcholines (lysoPCs) and acylcarnitines) using a synthetic agonist for RXR-Nurr1 heterodimer complex [36]. The approach to treat cerebral ischemia from the metabolism perspective is an important methodological consideration that differs from the current treatment methods that are based on mitigating the etiology of stroke by preventing or opening up the vascular occlusion. Unfortunately, only few studies have reported on the treatment of cerebral ischemia from the metabolism perspective, due to incomplete understanding of the metabolic changes occurring in the ischemic brain.

## 3. Metabolite Analytical Methods and Data Interpretation Platforms for Cerebral Ischemia

Metabolic profiling reflects the phenotype of biological changes in cellular mechanisms [37,38,39]. A metabolic profile can be analyzed with samples from various sources, including biological fluids, cells, tissues, and whole organisms (e.g., plants and microorganisms) [37]. In the case of cerebral ischemia, this can be done using microdialysis, nuclear magnetic resonance (NMR), and MS [13,14,40,41]. Microdialysis is used for continuous measurement of molecules, including neurotransmitters, hormones, and metabolites in the extracellular fluid coupled with liquid chromatography [42,43]. NMR has the advantage of determining structures of organic metabolites and quantifying a broad range of metabolites, but has a relatively higher detection limit (>~1 nmol) than MS and cannot analyze NMR-inactive metabolites [44]. MS has a high sensitivity and accuracy for quantification of metabolites and is generally performed with liquid chromatography (LC), gas chromatography (GC), and capillary electrophoresis (CE) for separation [45].

Metabolic data should be analyzed with appropriate statistical methods for selecting biomarkers and finding correlations between variables [46]. Basically, univariate statistics are used for a single variable, and multivariate statistics are used for more than two variables. Univariate statistics for parametric tests (*t*-test and analysis of variance (ANOVA)) are applied to normal (Gaussian) distributed datasets, while nonparametric tests (Kruskal–Wallis and Mann–Whitney *U* test) are applied to skewed or jagged datasets [47,48]. Generally, because data obtained from cerebral ischemia patients and animal models involve multiple covariances, multivariate statistical analysis is used to discriminate biomarkers and investigate correlations between the etiological evaluation as independent variables and metabolites as dependent variables [49]. Unsupervised methods of multivariate statistical analysis include principal component analysis (PCA), self-organizing maps, hierarchical clustering, and K-means. These methods reduce the dimensionality of data and visualize clusters (classifications) based on data similarity among samples [50,51,52]. Supervised methods of multivariate statistical analysis are used to analyze the relationship among a specific etiological evaluation and metabolites. The supervised multivariate statistical analyses consist of methods for regression and classification. Partial least squares analysis (PLS) is used for regression between response and predictor variables [53]. Orthogonal projections to latent structures (OPLS) is an advanced method of PLS by applying orthogonal signal correction filter for removing uncorrelated variables and reducing model complexity [54]. Classification is normally carried out by classifiers such as discrimination analysis (DA) [55], support vector machine (SVM), K nearest neighbor classification (K-NN), and soft-independent modeling of class analogies (SIMCA) [56], which are performed with each different algorithm.

Bioinformatic tools including metabolomics enable researchers to understand the complexity of cerebral ischemic events. Various kinds of bioinformatic tools are being developed for comprehensive analysis and integration of multi-omics [46]. MetaboAnalyst 3.0 [57], 3Omics [58], Integrated Molecular Pathway Level Analysis (IMPaLA) [59], integrOmics [60], MetScape [61], Paintomics [62], PathVisio [63], and Ingenuity Pathway Analysis (IPA) [64] have been used to interrogate the relationships among metabolites (and also with other omics) and biological functions. These approaches facilitate the identification of new biomarkers and biological functions and can aid in the understanding of physiologic and pathologic states. The event of cerebral ischemia likely triggers a multifaceted cascade of metabolic and biochemical events through metabolic changes caused by brain damage as detected in clinical samples and animal models [13,14,65]. These findings accentuate the need for advanced bioinformatic cellular metabolic analyses.

## 4. Metabolic and Cellular Analysis in Patients with Cerebral Ischemia

Various classes of small metabolites in plasma, serum, and urine have been analyzed and used to understand the biological changes occurring in patients with cerebral ischemia (Table 1) [46]. Alterations in metabolite levels have been reported, including organic acids, amino acids, free fatty acids, lipids, and low-density lipoprotein in the serum and plasma of cerebral ischemia patients. Basically, glucose metabolic pathways are highly affected by cerebral ischemia due to reduction in oxygen and nutrient availability [66]. Characteristically, glycolysis is changed from aerobic to anaerobic pathway in cerebral ischemia [67]. In addition, the tricarboxylic acid (TCA) cycle is suppressed by oxidative radicals due to the predominance of anaerobic glycolysis [68]. Moreover, the pentose-phosphate pathway is activated as an endogenous antioxidant mechanism by increasing nicotinamide adenine dinucleotide phosphate (NADPH)/nicotinamide adenine dinucleotide (NAD)+ ratio [69].

Time is a decisive factor in the diagnosis and treatment of cerebral ischemia, and often referred to as “time is brain” [15,20,24,30]. Time-dependent pathophysiological mechanisms of cerebral ischemia take place through several sequential steps [70]: (1) As a result of reduced blood flow and depletion of oxygen and nutrient delivery to brain tissue, energy depletion leads to excitotoxicity and peri-infarct depolarization within hours. (2) Proinflammatory cytokines generated by injured brain cells recruit macrophages and monocytes to the ischemic penumbra and trigger brain inflammation and oxidative stress within days. (3) The inflammatory condition and reactive oxygen species trigger necrosis and apoptosis of brain cells through mitochondrial and DNA damage for days and weeks. In addition, disruption of the blood–brain barrier (BBB) is one of the secondary events in brain injury and progression of cerebral ischemia [71,72]. BBB disruption is a major pathophysiological contributor to brain injury through the evolution of cerebral ischemia and is regulated by inflammatory modulators, oxidative damage, and altered regulation of adhesion molecules [17,73,74].

Several studies investigated metabolic changes in acute ischemic strokes of varying severities. In the SPOTRIAS (Specialized Programs of Translational Research in Acute Stroke Network) biomarker study, serum metabolites in the acute phase were measured within 2 h of iatrogenic MCAo in a rat model, and within 9 h of focal neurological deficit onset in ischemic stroke patients, with stroke severity stratified with the NIHSS [75]. Another study evaluated metabolic features of the serum of ischemic stroke patients with significant deficits (NIHSS ≥ 6) compared to healthy controls [76]. Although these studies analyzed different ranges of metabolites, both studies found that glycine, isoleucine, and lysine, which are highly related to inflammation, were present in low levels compared to healthy control subjects [75,76]. Changes related to membrane lipid metabolism were observed in serum of patients with MCAo, while changes in branched chain amino acids (BCAAs), leucine, isoleucine, and valine, were observed in cardioembolic stroke.

At the cellular level, excitotoxicity is the initial cellular mechanism of insult in cerebral ischemia [77]. This is triggered by failure in metabolic homeostasis and by secreted metabolites including glutamate, glycine, D-serine, and polyamines [78,79]. Excitotoxicity related metabolites bind to the N-methyl-D-aspartate (NMDA) receptor, which is a calcium ion-permeable, ligand-gated ion channel that triggers disturbances in calcium homeostasis in neurons [80]. In particular, the increase in the excitotoxic neurotransmitter glutamate disrupts the balance of glutamate/glutamine coupling, which is important for brain metabolic homeostasis between astrocytes and neurons [81]. At the neuronal level, calcium-related signaling cascades are processed with high speed with an approximate 20,000-fold difference in physiological concentration between the extracellular and intracellular environment (mM vs. 100 nM) [82]. This imbalance mainly affects the mitochondria and endoplasmic reticulum, which contain Ca^2+^ ion channels [82]. In particular, when Ca^2+^ ions accumulate massively in mitochondria, rendering them functionally defective, and leading to a reduction in adenosine triphosphate (ATP) production and the generation of reactive oxygen species (ROS) [83]. During severe mitochondrial dysfunction, neuronal cell death occurs via necrosis and oxidation of cellular components such as proteins, lipids, RNA, DNA, as well as via apoptosis with the activation of the caspase cascade [84]. Excitotoxic metabolic changes in cerebral ischemia have been noted in patients with ischemic stroke within seven days of the onset of stroke [85,86,87].

Oxidative stress in cerebral ischemia is caused by excessive production of oxygen derivatives and metabolic dysfunction [88]. Under normal conditions, oxygen derivatives, mainly ROS, are generated in the mitochondrial respiratory chain to produce ATP, and superoxide is converted to hydrogen peroxide, as a signaling messenger, by the enzyme superoxide dismutase. However, in cerebral ischemia, oxygen depletion leads to anaerobic glycolysis in cells, generating lactate as a final product, which in turn makes the cytosolic environment acidic [89]. Excessive protons convert oxygen into hydrogen peroxide and reactive hydroxyl radical. As a reflection of these metabolic changes in the brain, increased levels of formate, glycolate, lactate, and pyruvate have been reported in the plasma and urine of cerebral ischemia patients [40]. In addition, oxidative stress related to metabolic changes has been suggested in the profiles of amino acids, organic acids, free fatty acids, and polyamines in serum or plasma of patients [85,90,91,92].

An inflammatory response is also exacerbated by oxidative stress and blood–brain barrier damage in cerebral ischemia [15]. With the activation of inflammation, glial cells, neutrophils, monocytes, and lymphocytes, as well as levels of proinflammatory cytokines and inflammation related metabolites all increase [91]. Elevated BCAAs are known to be catabolized through ketogenic pathway to generate energy when nutrient availability is scarce in a robust inflammatory cascade and oxidative stress [93,94]. In patients with cerebral ischemia, levels of BCAAs are reported decreased by inflammation [75,85,86,91,95]. In addition, decrements in lysophosphatidylcholines and polyunsaturated free fatty acids, and increments in short-chain free fatty acids, short-chain acylcarnitines, and ceramides have been found in plasma from cerebral ischemia patients with inflammation and β-oxidation, but the metabolic mechanisms of these findings remain hypothetical [14,85,90,91,92,96].

Other pathophysiological and metabolic changes have also been suggested in the context of cerebral ischemia [40,65,76,96,97,98,99,100,101]. Increased levels of formate and glycolate and decreased levels of dimethylamine, glycine, hippurate, and methanol have been suggested to be related with hyperhomocysteinemia [40]. The activation of one-carbon metabolism, including folate cycle and methionine cycle, is believed to change the levels of folic acid, cysteine, S-adenosyl homocysteine, and oxidized glutathione, and these changes may represent biomarkers in sera of cerebral ischemia patients [97]. Elevated levels of C5:1 (tiglylcarnitine) have been suggested to relate to platelet dysfunction by peroxisome proliferator-activated receptor α (PPARα), and elevated levels of alanine are suggested to be related to neuronal autophagy by increasing carnosine content in blood samples of cerebral ischemia patients [99]. Altered levels of phosphorus-containing compounds, ceramides, amino acids, fatty acids, carbohydrates, cholines, polyamines, and membrane lipids have been suggested as biomarkers for cerebral ischemia, but the lack of detailed mechanistic explanation for these changes represents a limitation [65,76,85,96,98,100,101]. Moreover, there are discrepancies in reported patterns of analyzed metabolite levels among studies. These discrepancies derive from differences in sample sourcing and complexity in disease severity, onset time, race, diet, and age of patients. In addition, most studies provided insufficient data regarding ischemic stroke subtypes such as the Trial of Org 10,172 in Acute Stroke Treatment (TOAST) classification. Thus, standardization of the various critical factors for future studies is necessary to identify definitive metabolic biomarkers for cerebral ischemia as well as controlled standardized experiments employing well defined in vivo and in vitro models.

## 5. Metabolic and Cellular Analysis in Animal Models of Cerebral Ischemia

Various animal models of cerebral ischemia have been established to analyze underlying mechanisms and to develop therapies. These models are necessary and useful for several reasons: (1) They represent a controlled and standardized system, allowing highly reproducible analyses, (2) molecular and biochemical analyses are possible using brain tissue, (3) Time-dependent imaging analysis is feasible in animal models, and (4) perfusion and vasculature are not available in in vitro systems. Small animals can be challenging to handle and model for cerebral ischemia; therefore, larger animals are often used for this purpose [102]. In addition, infarct size is known to be affected by rodent strain [103]. For example, spontaneously hypertensive rats exhibit large infarcts with low variability, whereas Sprague–Dawley rats have relatively small infarcts with high variability with MCAo [104]. As expected, infarct volume correlates with mortality rate [105], and high variability reduces statistical power [106]. In the case of mouse strains, C57B1/6 mice are reported to generate relatively larger infarcts compared with Sv129 mice [107]. Animal models of cerebral ischemia include the intraluminal filament MCAo, direct MCA ligation by a subtemporal craniectomy approach, photothrombosis, endothelin-1, and the embolic stroke model [103,108,109,110]. Among these, the rodent intraluminal filament MCAo model is the most commonly used because of the high similarity with human cerebral ischemia, exhibits a penumbra, is highly reproducible, has a highly controllable reperfusion, and does not require craniectomy [103]. However, the MCAo model also has limitations such as hyper-/hypothermia issues, increased hemorrhage with certain filament types, and the fact that it is unsuitable for thrombolysis studies. In addition, transgenic and knockout mice have been developed and used for studies of genomic and proteomic mechanisms of cell injury, neuroprotection, and redox mechanisms of cerebral ischemia [111,112].

Metabolites in animal models have also been analyzed and used to understand the biological changes occurring in cerebral ischemia (Table 2). Changes in metabolite levels have been reported, including amino acids, free fatty acids, lipids, nucleotides, short peptides, phosphoethanolamine (PE), phosphoserine (PS), and phosphocholines (PCs), in the plasma, brain tissue, and cerebrospinal fluid (CSF) in MCAo rodent models. Similar to findings in patients, reductions in metabolites related to branched chain amino acids have also been observed in the rat MCAo model [75]. Moreover, we previously quantified free fatty acids from the plasma and brain tissue of MCAo rats using GC–MS and found notable changes in the levels of polyunsaturated fatty acids (PUFAs) [14]. PUFAs are highly related to oxidative stress [113] and inflammation during cerebral ischemia [114]. We found decreased levels of PUFAs in plasma, such as linoleic acid and arachidonic acid, consistent with a similar trend in a previous study in human subjects [13]. Changes in polyamine levels in MCAo rat brain tissue have also been reported and suggested to be related to oxidative stress and the polyamine interconversion pathway [65]. Additionally, changes in the levels of *N*-acetyl-aspartate (NAA) serve as an important biomarker for cerebral ischemia [115]. NAA is a source for acetyl groups and is related to neurotransmitter metabolism [116]. Moreover, NAA is a marker of neuronal integrity because it is synthesized in the neuronal mitochondrial membrane [117]. In addition, NADP+/NADPH and NAD+/NADH represent redox status, energy-dependent process, and energy phosphate stores [118]. Future metabolomic studies should be focused on these important metabolites.

We have analyzed metabolic changes in the rat MCAo model using a bioinformatics tool to understand patterns of biological phenotypes. 2,3,5-Triphenyltetrazolium chloride (TTC) staining image depicts the infarct region in the ipsilateral cortex and striatum of MCAo rats, which is similar to the coronal T2 weighted image of a patient with MCA territory stroke (Figure 1). Metabolites, including 29 amino acids (AAs), 21 organic acids (OAs), and nine polyamines (PAs), in brain tissue lysates from MCAo rats were quantified using gas chromatography–mass spectrometry (GC–MS) (Table 3) [65]. Representative selected-ion monitoring (SIM) chromatograms showed changes in the levels of AAs, OAs, and PAs in MCAo brain tissue compared to control brain tissue (Figure 2). The levels and normalized values of AAs, OAs, and PAs are clearly revealed with increased and decreased patterns in metabolic network. Specifically, in the AAs profile, an increase in leucine, isoleucine, methionine, phenylalanine, 4-hydroxyproline, lysine, tyrosine, and 3,4-dihydroxyphenyl-alanine (DOPA) levels, and a decrease in gamma-aminobutyric acid (GABA), aspartic acid, glutamic acid, and glutamine levels were observed. In OAs profile, an increase in 3-hydroxybutyric acid, pyruvic acid, glycolic acid, malonic acid, and citric acid and a decrease in acetoacetic acid, fumaric acid, oxaloacetic acid, and malic acid were observed. In PAs profile, increases in putrescine, cadaverine, *N*^1^-acetylspermidine, and spermidine levels were observed. However, there are limitations in interpreting comprehensively the relationship between alterations in metabolite levels and complex cerebral ischemia events. Thus, the quantified metabolites were analyzed based on their biological function using IPA (Ingenuity Systems, http://www.ingenuity.com) [64], which is a web-based bioinformatics software used to identify biological functions. The visualized network showed a clear relationship between metabolites and related biological functions, including oxidative stress response, inflammatory response, lipid peroxidation, mitochondrial transmembrane potential, and apoptosis (Figure 3A). Network prediction revealed activation of the oxidative stress response, inflammatory response, lipid peroxidation, apoptosis, and a reduction in mitochondrial transmembrane potential (Figure 3B). Thus, in silico predictions based on metabolic network analysis are consistent with the physiochemical events occurring in cerebral ischemia.

## 6. Metabolic and Cellular Analysis in in Vitro Models of Cerebral Ischemia

There are some differences between animal models of cerebral ischemia and patients with stroke. Although these models are a good reflection of human cerebral ischemia and have been crucial in our understanding of the pathophysiologic events in cerebral ischemia, there are genetic differences between animals and humans. For example, the human and rat genomes differ by 10% [121]. As a result, differences were found in blood–brain barrier function, excitotoxicity, and inflammatory responses [121,122,123]. These differences call for the utilization of in vitro systems to supplement observations in animal models of the cerebral ischemia. Therefore, the development of in vitro based methods is useful for an initial rapid evaluation and supplementation of experiments in animal models. In addition, animal models of cerebral ischemia can be complicated and are often time-consuming, especially with the need for validation of ischemic pathophysiological condition. Therefore, the development of in vitro based methods is useful for initial rapid evaluation and as supplementary to experiments in animal models.

There are three representative in vitro ischemic models: oxygen–glucose deprivation (OGD), excitotoxicity, and oxidative phosphorylation blocking methods. First, in vitro cerebral ischemia models are usually based on OGD using chemicals or enzymes that induce glucose deprivation and hypoxic chambers for oxygen deprivation [124]. Second, studies employing the excitotoxicity model, which is part of ischemic damage, employ *N*-Methyl-d-aspartate (NMDA) or glutamate [125,126,127]. Third, oxidative phosphorylation blocking methods, such as using sodium azide, rotenone, and antimycin, which inhibit the electron transport chain [124].

We previously analyzed polyamine profiles in the rat MCAo model and in the in vitro human neuroblastoma SH-SY5Y cell line OGD system using GC–MS [65]. Levels of putrescine, cadaverine, *N*^1^-acetylspermidine, and spermidine were significantly higher in brain tissue in the rat model, while putrescine, *N*^1^-acetylspermidine, *N*^8^-acetylspermidine, spermidine, *N**^1^*-acetylspermine, and spermine were significantly higher in the OGD system. In particular, spermine levels were elevated only in in vitro models. Even though there are some differences between animal models and in vitro systems, in vitro ischemic models represent a helpful means to support the metabolic and cellular mechanisms of pathophysiological conditions in ischemia.

## 7. Conclusions

Here, we reviewed state-of-the-art techniques and analyses of metabolic and cellular changes to study cerebral ischemia. In metabolic analyses involving patients, discrepancies have been observed in patterns of up- or downregulation of metabolites due to limitations in standardizing disease severity, onset time, race, diet, age, and other patient characteristics. Moreover, a detailed etiological evaluation is also necessary for standardization. In future studies, screening for biomarkers in human subjects should be accompanied by a precise evaluation of clinical variables. Animal models for cerebral ischemia have the advantage of reproducibility, biochemical analysis, time-dependent analysis, and ability to perform perfusion experiments, compared to human subjects and in vitro models. However, there are some limitations in animal models, such as genomic differences between humans and rodents as well as differences in biological phenotypes, plus the time and labor needed to generate and maintain animal models.

Conventional detection methods are limited and cannot comprehensively portray and identify the complex relationships between metabolic changes and cellular and biological functions in the context of cerebral ischemia. Metabolomics has the unique property of reflecting the endpoint phenotype, and a combination of metabolomics and bioinformatics can facilitate the understanding of cellular and system changes in cerebral ischemia. In addition, various models to mimic actual conditions of cerebral ischemia in patients have been developed using animals and in vitro systems. Future studies in cerebral ischemia should integrate multi-omics and a multidisciplinary use of omics methods with advanced validation models. The omics integration will be a more comprehensive approach to study cerebral ischemia and can aid in the discovery of novel biomarkers, new pathophysiological mechanisms, and innovative effective treatments for cerebral ischemia.

## Figures and Tables

**Figure 1 cells-09-01630-f001:**
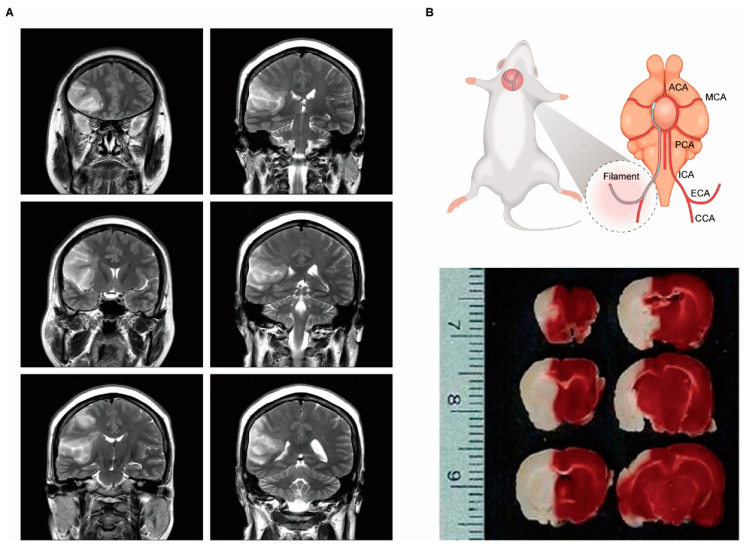
(**A**) Coronal T2 weighted images. High signal intensity indicates cerebral infarction in the middle cerebral artery (MCA) territory from a patient with occlusion of M1 of MCA and mild stenosis of the proximal cervical internal carotid artery with fibrofatty plaque. Coronal T2 weighted magnetic resonance images were acquired using a clinical 3 T scanner (GE Healthcare, Signa pioneer, USA) with a 24-channel coil, and the following parameters: repetition time (TR) 5745 ms; echo time (TE) 116 ms; flip angle of 142°; matrix size 416 × 416; and 5.0  mm slice thickness. This study was approved by the Scientific-Ethical Review Board of Ajou University Medical Center (AJIRB-MED-EXP-20-044). (**B**) Schematic diagram of the surgical procedure of MCAo model, and cerebral coronal sections (2 mm thick) of MCAo lesioned rat brains at 24 h were stained with 2, 3, 5-Triphenyltetrazolium chloride (TTC). Anterior communication artery (ACA); posterior cerebral artery (PCA); internal carotid artery (ICA); external carotid artery (ECA); common carotid artery (CCA). TTC-stained image is reproduced from our previous study, Copyright© 2009 with permission from Elsevier [14].

**Figure 2 cells-09-01630-f002:**
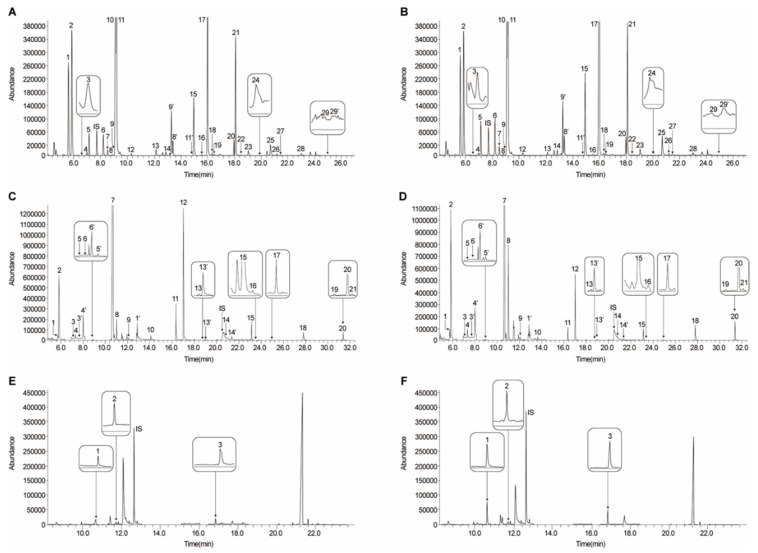
GC–MS chromatograms of 29 AAs, 21 OAs, and 9 PAs. SIM chromatograms of control and 5 days post MCAo rat brain. AAs from control (**A**) and MCAo rat brain (**B**) SIM chromatograms: 1 = alanine; 2 = glycine; 3 = α-aminobutyric acid; 4 = valine; 5 = β-aminoisobutyric acid; IS = norvaline; 6 = leucine; 7 = isoleucine; 8 = threonine; 9 = serine; 10 = proline; 11 = γ-aminobutyric acid (GABA); 12 = pipecolic acid; 13 = pyroglutamic acid; 14 = methionine; 9′ = serine; 8′ = threonine; 11′ = GABA; 15 = phenylalanine; 16 = cysteine; 17 = aspartic acid; 18 = *N*-methyl-dl-aspartic acid; 19 = 4–hydroxyproline; 20 = homocysteine; 21 = glutamic acid; 22 = asparagine; 23 = ornithine; 24 = α-aminoadipic acid; 25 = glutamine; 26 = lysine; 27 = histidine; 28 = tyrosine; 29 = DOPA; 29′ = DOPA. OAs from control (**C**) and MCAo rat brain (**D**) SIM chromatograms: 1 = 3-hydroxybutyric acid; 2 = pyruvic acid; 3 = α-ketoisovaleric acid; 4 = acetoacetic acid; 3′ = α-ketoisovaleric acid; 4′ = acetoacetic acid; 5 = α-ketoisocaproic acid; 6= α-keto-β-methylvaleric acid; 6′= α-keto-β-methylvaleric acid; 5′= α-ketoisocaproic acid; 7 = lactic acid; 8 = glycolic acid; 9 = 2-hydroxybutyric acid; 1′ = 3-hydroxybutyric acid; 10 = malonic acid; 11 = succinic acid; 12 = fumaric acid; 13 = oxaloacetic acid; 13′ = oxaloacetic acid; IS = 3,4-dimethoxybenzoic acid; 14 = α-ketoglutaric acid; 14′ = α-ketoglutaric acid; 15 = 4-hydroxyphenylacetic acid; 16 = malic acid; 17 = 2-hydroxyglutaric acid; 18 = *cis*-aconitic acid; 19 = 4-hydroxyphenyllactic acid; 20 = citric acid; 21 = isocitric acid. PAs from control (**E**) and MCAo rat brain (**F**) SIM chromatograms: 1 = putrescine; 2 = cadaverine; 3 = spermidine. PAs data is reproduced from our pervious study, Copyright © 2016 [65].

**Figure 3 cells-09-01630-f003:**
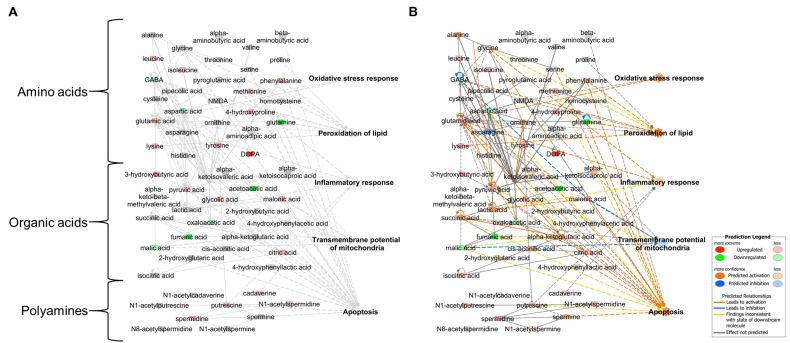
Functional analysis of metabolomic network of MCAo rat brain tissue using IPA. Metabolomic network (**A**) and metabolomic network with prediction (**B**). Fold change ± 1.2 was used as cut-off value. Red and green areas indicate up- and downregulated metabolites, respectively. Orange and blue areas indicate prediction as activation and inhibition, respectively. Prediction Legends are originated from Ingenuity Systems (http://www.ingenuity.com).

**Table 1 cells-09-01630-t001:** Summary of metabolites associated with cerebral ischemia in patients.

Increased Metabolic Biomarkers	Decreased Metabolic Biomarkers	Subjects and Specimens	Patient Groups	Analysis Method	Suggested Biological Changes	Reference
Formate, glycolate, lactate, pyruvate	Glutamine, lipid (CH_2_CH_2_C = C), methanol, valine, VLDL/LDL CH_3_	Plasma from patients with cerebral infarction	Suspected ischemic event within 72 h of onset	Proton (^1^H) -NMR	Metabolic acidosis and hyperhomocysteinemia	[40]
-	Citrate, creatinine, dimethylamine, glycine, hippurate	Urine from patients with cerebral infarction				
Cysteine, hydroxyeicosatetraenoic acid, hydroxyoctadecadienoic acid, oxidized glutathione, S-adenosylhomocysteine	Adenosine, aldosterone, betanin, desoxycortone, folic acid, sucrose-6-phosphate, tetrahydrofolic acid	Serum from patients with cerebral infarction	Acute onset of neurological deficit lasting less than 6 h	Ultrahigh pressure LC-TOF- MS	One-carbon cycle metabolism	[97]
Glucose, *N*-carbamoyl-beta alanine	Asparagine, carnitine, *cis*/*trans*-hydroxyproline, creatinine, cysteamine, glycine, histidine, isoleucine, leucine, lysine, methionine, proline, threonine, uridine, valine 5′-adenosylhomocysteine	Plasma from patients with acute stroke (neuroimaging evidence of infarction, or symptom duration >24 h)	Mild and severe stroke groups classified with SPOTRIAS ^a^ biomarker cohort	Hydrophilic interaction chromatography MS/MS	Branched chain amino acids metabolism	[75]
Inorganic phosphate	Phosphatidylcholine, phosphoethanolamine, sphingomyelin	Serum from patients with ischemic stroke	Patients with or without thrombolytic therapy	Phosphorus-31 (^31^P)-NMR	Increment of phosphorus-containing compounds	[98]
Carnitine, creatinine, glutamine, hypoxanthine, kynurenine, *N*-acetylneuraminic acid, palmitoylcarnitine, phenylalanine, proline, sphingosine 1-phosphate, tyrosine, uric acid	Citric acid, isoleucine, lysophosphatidylcholines (14:0, 16:0, 18:0, 18:1, 18:2, 20:1), tryptophan, valine	Serum from patients with post-stroke patients	post-stroke cognitively (MoCA ^b^ < 24) or non-cognitively impaired (MoCA ≥ 24) patients	Ultra-high-performance LC- quadrupole time-of-flight (QTOF)-MS	Branched chain amino acids metabolism, stroke-induced inflammation, increased neurotoxicity, bioenergetic homeostasis, increased oxidative stress, cell apoptosis, and other function	[91]
Aspartic acid	Cholesterol, creatinine, isoleucine, linoleic acid, oleic acid, palmitic acid, phenylalanine, proline, pyroglutamate, serine, stearic acid, valine	Plasma from depressed stroke patients	Patients with or without post-stroke depression	GC–MS	Oxidative stress, mitochondrial dysfunction, energy homeostasis, and inflammation	[92]
Alanine, citrulline, 3-hydroxyisovalerylcarnitine (C5–OH), C5–OH/C0, C5–OH/C8	-	Dried blood spot from patients with cerebral infarction	Patients with cerebral infarction and intracerebral hemorrhage	Direct injection LC-MS/MS	Attenuation of neuronal autophagy, mitochondrial damage, apoptosis, and platelet dysfunction	[99]
Isoleucine, leucine, valine	-	Plasma from patients with cardiovascular disease	Patients with cardiovascular disease, including ischemic and hemorrhagic stroke	LC-MS/MS	Branched chain amino acids metabolism	[95]
Short- and medium-chain acylcarnitines	-	Plasma from patients with cardiovascular disease	Patients with cardiovascular disease, including stroke	LC-MS/MS	β- oxidation, altered mitochondrial metabolism, reactive oxygen species, and inflammation	[90]
Glutamic acid	Glutamine, glutamine/glutamic acid ratio	Plasma from patients with cardiovascular disease	Patients with cardiovascular disease, including stroke	LC-MS/MS	Glutamate-cycling pathway	[100]
Ceramides (16:0, 22:0, 24:0, and 24:1)	-	Plasma from patients with cardiovascular disease	Patients with cardiovascular disease, including stroke	LC-MS/MS	Insulin resistance and inflammation	[96]
Homocysteine sulfinic acid	Cadaverine, L-lysine, L-valine, nicotinamide, N6-acetyl-Ll-ysine, S-(2-methylpropionyl)-dihydrolipoamide-E, ubiquinone, 2-oxoglutarate, 5-aminopentanoate	Serum from patients with thrombotic stroke	Patients with thrombotic stroke	Ultra performance LC-QTOF-MS	Catabolism process of lysine, excitotoxity, oxidative stress, branched chain amino acids metabolism, and inflammation	[85]
Acetylcarnitine, betaine, carnitine, galactose, lysophosphatidylethanolamines (18:2, 20:2, 20:4, 20:5), L-isoleucyl-L-proline, mannose	Alanine, aspartate, glycine, isoleucine, lysine, lysophosphatidylcholine (16:0), ornithine, phosphatidic acid (18:3/0:0), phosphatidylcholine (1:0/16:0, 5:0/5:0), phosphatidylinositol (22:2/0:0), proline, serine, threonine, tricarballylic acid, trihydroxy palmitic acid	Serum from patients with acute ischemic stroke	Patients with ≤ 9 h after stroke onset, and stroke localization in the area of the middle cerebral artery	GC–MS, ultrafast liquid chromatography coupled with ion trap time-of-flight mass spectrometry	Amino acid, fatty acid, carbohydrate, choline, and membrane lipids metabolism	[76]
Glutamate, lactate, phenylalanine/tyrosine ratio, tryptophan	Alanine, citrate, erythronic acid, glycine, H-purine, isoleucine, leucine, lysine, methionine, proline, serine, tyrosine, urea	Serum from patients with acute ischemic stroke	Patients with ischemic stroke symptoms within seven days of the onset	GC–MS	Branched chain amino acids metabolism and excitotoxity	[86]
Diacylglycerol (38:6), lysophosphatidylcholines (20:4, 20:5, 22:6, 24:0), triacylglycerols (52:5, 54:3, 54:4, 54:5, 56:5)	Free fatty acid (16:1), glucosylceramide (38:2), phosphatidylethanolamine (35:2)	Plasma from patients with lacunar infarction	Patients with lacunar infarction had clinical presentation and brain neuroimaging evidence of infarct size ≤1.5 cm at a typical location	2D (normal phase/reverse phase) LC-QTOF-MS	Hypertriglyceridemia	[101]
Docosatrienoic acid, phytosphingosine, sphinganine, tetradecanedioic acid	Glutamine, lysophosphatidylethanolamines [0:0/22:0], pyroglutamic acid, 2-Ketobutyric acid	Plasma from patients with ischemic stroke	Patients with cerebral infarction and intracerebral hemorrhage	UPLC-QTOF-MS	Excitotoxity, apoptosis, and energy metabolism	[87]

^a^ SPOTRIAS: Specialized programs of translational research in acute stroke. ^b^ MoCA: Montreal cognitive assessment. Table is modified from Au, A. Metabolomics and Lipidomics of Ischemic Stroke. *Adv Clin Chem*
**2018**, *85*, 31–69, Copyright © 2018 with permission from Elsevier [46].

**Table 2 cells-09-01630-t002:** Summary of metabolites associated with MCAo in the rat model.

Increased Metabolic Biomarkers	Decreased Metabolic Biomarkers	Subjects and Specimens	Analysis Method	Suggested Biological Changes	Reference
Xanthosine, carnosine, glutamate	leucine, isoleucine, valine, phenylalanine niacinamide	Plasma and cerebrospinal fluid from MCAo rat model	Hydrophilic interaction chromatography MS/MS	Branched chain amino acids metabolism	[75]
Caproic acid (C6:0), Caprylic acid (C8:0), Decenoic acid(C10:1), Capric acid (C10:0), Lauric acid (C12:0), Myristoleic acid (C14:1), Myristic acid (C14:0), γ-Linolenic acid (C18:3n6), Eicosapentaenoic acid (C20:5n3), Docosahexaenoic acid (C22:6n3), Docsapentaenoic acid (C22:5n3), Erucic acid (C22:1), Nervonic acid (C24:1), Lignoceric acid (C24:0), Hexacosanoic acid (C26:0)	Palmitic acid (C16:0), Linoleic acid (C18:2n6), Arachidonic acid (C20:4n6), Eicosenoic acid (C20:1)	Plasma from MCAo rat model	GC–MS	Inflammation and oxidative stress	[14]
Myristic acid (C14:0), Linoleic acid (C18:2n6), Stearic acid (C18:0), Eicosenoic acid (C20:1), Eicosanoic acid (C20:0), Docsapentaenoic acid (C22:5n3), Erucic acid (C22:1), Behenic acid (C22:0)	Palmitic acid (C16:0), Oleic acid (C18:1)	Brain tissue from MCAo rat model			
Putrescine, Cadaverine, *N*^1^-acetylspermidine, Spermidine	-	Brain tissue from MCAo rat model	GC–MS	Oxidative stress and polyamine interconversion pathway	[65]
Lactate, Glutamate, Glycerol	Glucose	Cerebrospinal fluid from MCAo mouse model	Microdialysis	Energy metabolism and cellular damage	[119]
Leucine, Isoleucine, Valine, 3-Hydroxyisobutyrate, 3-Hydroxybutyrate, Lactate, β-Alanine, Alanine, Lysine, Glutamine, Succinate, Methionine, Ethanolamine, Choline, sn-Glycero-3-phosphocholine, Taurine, Glucose, Glycine, Threonine, Uracil, Cytidine, Fumarate, Tyrosine, Anserine, Phenylalanine,	GABA, Acetate, N-acetyl-Aspartate, Glutamate, Glutathion, Aspartate, Trimethylamine, Creatine, Malonate, Phosphocholine, Ascorbate, myo-Inositol, Inosine, UDP-galactose, UDP-glucose, Uridine, Guanosine, AMP, ADP, ATP, Carnosine, Nicotinamide, Oxypurinol, Hypoxanthine, NADH	Brain tissue from MCAo rat model	^1^H NMR	Oxidative stress, inflammation, energy metabolism, amino acid metabolism, and neuronal and glial integrity	[115]
Cholic acid, Pseudouridine/Uridine, pyruvate, Taurine, Lactate, Glutamine, Alanine, taurine, Phenylalanine, Tryptophan, Histidine Norepinephrine, 5-hydroxyindoleacetic acid, LysoPE(24:0), cytidine, creatine, Palmitoyl-l-carnitine, *N*-stearoylserine	Gly-Gln-Leu, Gly-Cys-Ala-Phe, PC(14:0/0:0), PE(17:0/0:0), PS(20:0/0:0), PC(14:1), LysoPE(0:0/16:0), LysoPE(20:1), LysoPE(0:0/24:1(15Z)), LysoPE(20:0/0:0), acetone	Plasma from MCAo rat model	UPLC-Q/TOF-MS	Amino acid metabolism, energy metabolism, lipid metabolism	[120]

**Table 3 cells-09-01630-t003:** The amount of 29 AAs, 21 OAs, and 9 PAs in MCAo rat brains.

	AAs	Amount (ng, Mean ± SD) in Rat Brain Tissue (mg)			OAs	Amount (ng, Mean ± SD) in Rat Brain Tissue (mg)			PAs	Amount (ng, Mean ± SD) in Rat Brain Tissue (mg)		
No.	Analyte	Control	MCAo	Ratio	Analyte	Control	MCAo	Ratio	Analyte	Control	MCAo	Ratio
1	Alanine	191.0 ± 92.1	195.1 ± 119.2 (0.47) ^a^	1.02 ^b^	3-Hydroxybutyric acid	0.010 ± 0.004	0.02 ± 0.01 (0.24)	1.87	*N*^1^-Acetylputrescine ^b^	48.78 ± 1.99	49.51 ± 3.16 (0.812)	1.01
2	Glycine	145.5 ± 93.0	151.0 ± 103.5 (0.45)	1.04	Pyruvic acid	0.3 ± 0.1	0.3 ± 0.2 (0.11)	1.24	*N*^1^-Acetylcadaverine	60.45 ± 2.55	61.58 ± 4.23 (0.746)	1.02
3	α-Aminobutyric acid	10.1 ± 0.8	10.2 ± 0.5 (0.40)	1.01	α-Ketoisovaleric acid	0.005 ± 0.001	0.0048 ± 0.0003 (0.14)	0.91	Putrescine	9.63 ± 0.44	12.88 ± 0.86 (0.000001)	1.34
4	Valine	108.4 ± 20.8	125.6 ± 20.0 (0.05)	1.16	Acetoacetic acid	0.05 ± 0.02	0.03 ± 0.01 (0.30)	0.25	Cadaverine	6.08 ± 0.54	7.34 ± 0.96 (0.004)	1.21
5	β-Aminoisobutyric acid	9.3 ± 0.4	9.4 ± 0.6 (0.30)	1.01	α-Ketoisocaproic acid	0.031 ± 0.007	0.04 ± 0.01 (0.04)	1.19	*N*^1^-Acetylspermidine	25.66 ± 1.85	32.81 ± 4.94 (0.005)	1.28
6	Leucine	133.3 ± 36.2	181.3 ± 57.3 (0.02)	1.36	α-Keto-β-methylvaleric acid	N.D	N.D		*N*^8^-Acetylspermidine ^c^	43.41 ± 1.95	45.36 ± 2.79 (0.207)	1.04
7	Isoleucine	118.1 ± 42.2	157.7 ± 46.1 (0.04)	1.34	Lactic acid	3.9 ± 0.5	4.0 ± 0.5 (0.11)	1.04	Spermidine	25.07 ± 2.00	37.91 ± 4.66 (0.0003)	1.51
8	Threonine	130.3 ± 86.7	139.1 ± 94.8 (0.35)	1.07	Glycolic acid	0.11 ± 0.05	0.2 ± 0.1 (0.49)	1.45	*N*^1^-Acetylspermine	34.38 ± 5.91	32.65 ± 4.30 (0.801)	0.95
9	Serine	291.3 ± 216.9	338.4 ± 259.8 (0.36)	1.16	2-Hydroxybutyric acid	N.D	N.D		Spermine	103.92 ± 4.31	105.05 ± 6.66 (0.896)	1.01
10	Proline	134.6 ± 62.3	142.1 ± 99.7 (0.42)	1.06	Malonic acid	0.011 ± 0.002	0.017 ± 0.007 (0.002)	1.52				
11	GABA	1035.5 ± 131.4	663.2 ± 306.3 (0.002)	0.64	Succinic acid	0.03 ± 0.02	0.04 ± 0.02 (0.18)	1.12				
12	Pipecolic acid	9.9 ± 1.7	11.0 ± 3.2 (0.19)	1.11	Fumaric acid	0.008 ± 0.002	0.002 ± 0.001 (0.13)	0.31				
13	Pyroglutamic acid	12.5 ± 1.9	13.4 ± 2.1 (0.19)	1.07	Oxaloacetic acid	0.06 ± 0.04	0.042 ± 0.007 (0.06)	0.71				
14	Methionine	308.1 ± 87.8	420.5 ± 201.6 (0.07)	1.36	α-Ketoglutaric acid	0.026 ± 0.001	0.026 ± 0.002 (0.11)	1.01				
15	Phenylalanine	8.5 ± 1.2	11.0 ± 4.9 (0.07)	1.29	4-Hydroxyphenylacetic acid	0.017 ± 0.001	0.017 ± 0.001 (0.14)	1.00				
16	Cysteine	29.0 ± 1.6	30.0 ± 2.5 (0.16)	1.03	Malic acid	0.06 ± 0.01	0.046 ± 0.008 (0.03)	0.73				
17	Aspartic acid	439.7 ± 357.0	298.3 ± 252.5 (0.18)	0.68	2-Hydroxyglutaric acid	0.023 ± 0.001	0.024 ± 0.001 (0.13)	1.02				
18	*N*-Methyl-dl-aspartic acid	12.8 ± 0.8	12.8 ± 0.7 (0.49)	1.00	*cis*-Aconitic acid	0.025 ± 0.001	0.025 ± 0.001 (0.21)	1.01				
19	4-Hydroxyproline	10.9 ± 16.5	19.0 ± 21.9 (0.19)	1.74	4-Hydroxyphenyllactic acid	0.0171 ± 0.0007	0.017 ± 0.001 (0.15)	1.02				
20	Homocysteine	23.5 ± 1.2	23.6 ± 1.4 (0.29)	1.00	Citric acid	0.05 ± 0.01	0.07 ± 0.02 (0.32)	1.46				
21	Glutamic acid	142.6 ± 87.5	185.9 ± 138.4 (0.21)	1.30	Isocitric acid	0.022 ± 0.001	0.023 ± 0.001 (0.32)	1.04				
22	Asparagine	59.8 ± 4.8	70.5 ± 17.2 (0.03)	1.18								
23	Ornithine	54.6 ± 10.8	59.8 ± 13.3 (0.18)	1.10								
24	α-Aminoadipic acid	43.2 ± 2.1	43.5 ± 2.5 (0.44)	1.01								
25	Glutamine	722.8 ± 605.4	210.8 ± 459.3 (0.03)	0.29								
26	Lysine	98.8 ± 14.8	119.3 ± 30.0 (0.04)	1.21								
27	Histidine	207.3 ± 87.7	215.1 ± 101.2 (0.43)	1.04								
28	Tyrosine	40.8 ± 4.5	60.0 ± 38.0 (0.07)	1.47								
29	DOPA	2.8 ± 5.4	9.6 ± 15.2 (0.18)	3.43								

^a^ Student t-test comparing the mean values of the MCAo group with those of the control group. ^b^ Ratio of analyte levels in MCAo group to corresponding mean values in the control group. N.D: Not detected. PAs data is reproduced from our pervious study, Copyright © 2016 [65]. ^b^
*N*^1^-Acetyl: Acetylation at N1-position of chemical. ^c^
*N*^8^-Acetylspermidine: Acetylation on the terminal nitrogen adjacent to the 4-carbon chain of spermidine.

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
