# Peer review of "Metabolome Changes in Cerebral Ischemia"

_cells, 2020, doi:10.3390/cells9071630_

Round 1

Reviewer 1 Report

The manuscript entitled “Metabolic and cellular mechanisms of cerebral ischemia” by Tae Hwan Shin et. al. aims to review the metabolic and cellular changes in patients and animal models with cerebral ischemia. The paper is well written. The content of the article is informative, and the topic is the hot spot of current research on stroke. The manuscript will be improved by carefully editing.

There are several specific concerns with the manuscript that need to be addressed.

  1. It will be helpful to the review article for the author to briefly introduce the pathological mechanisms of cerebral ischemia, and emphasize that the changes in metabolism and cellular levels elicited by cerebral ischemia are the early events, occur within minutes after cerebral ischemia (Pathobiology of ischemic stroke: an integrated review. Trends Neurosci 22, 391-397), which in turn trigger or exacerbate a cascade of secondary brain injury, including neuronal cell death, neuroinflammation, blood-brain barrier disruption, etc.

  1. For the introduction, the author should emphasize the importance and significance of the intervention and treatment of stroke from the perspective of metabolism.

  1. It will be very helpful for the author to briefly introduce the change of the major glucose metabolic pathways including glycolysis, TCA cycle, Pentose-phosphate pathway before and after ischemic stroke, which lead to the metabolites change.  

  1. It will be also very helpful for the author to introduce the change of the metabolites in blood after cerebral ischemia.

  1. Glutamate/Glutamine coupling between astrocyte and neurons are very important for brain metabolism homeostasis. Accumulating evidence suggested that cerebral ischemia leads to neuronal and glial metabolic uncoupling, in which there is an acute increase in extracellular glutamate levels. It will be helpful for the author to include this information in this review article.

  1. The Microdialysis coupled with LC/MS or HPLC are the classical investigation methods for the analysis of extracellular metabolites after a variety of brain injuries, including cerebral ischemia. The author should also introduce this method.

  1. For the “Summary of metabolites associated with MCAo in the rat model”, maybe some information regarding the metabolite change in the rat stroke model is still missing. For example, some important metabolites such as N-acetyl-Aspartate, NAD+, NADH, NADP+, NADPH, etc, the most commonly reported metabolites in stroke studies, had not been introduced in the present review article.  

Author Response

Answers to the comments

Reviewer # 1

The manuscript entitled “Metabolic and cellular mechanisms of cerebral ischemia” by Tae Hwan Shin et. al. aims to review the metabolic and cellular changes in patients and animal models with cerebral ischemia. The paper is well written. The content of the article is informative, and the topic is the hot spot of current research on stroke. The manuscript will be improved by carefully editing.

There are several specific concerns with the manuscript that need to be addressed.

COMMENT 1: It will be helpful to the review article for the author to briefly introduce the pathological mechanisms of cerebral ischemia, and emphasize that the changes in metabolism and cellular levels elicited by cerebral ischemia are the early events, occur within minutes after cerebral ischemia (Pathobiology of ischemic stroke: an integrated review. Trends Neurosci 22, 391-397), which in turn trigger or exacerbate a cascade of secondary brain injury, including neuronal cell death, neuroinflammation, blood-brain barrier disruption, etc.

RESPONSE: Thank you for your suggestion. We added your suggestion to page 4 as follows:

Time-dependent pathophysiological mechanisms of cerebral ischemia take place through several sequential steps as follows [70]: (i) As a result of reduced blood flow and depletion of oxygen and nutrient delivery to brain tissue, energy depletion leads to excitotoxicity and peri-infarct depolarization within hours; (ii) Proinflammatory cytokines generated by injured brain cells recruit macrophages and monocytes to the ischemic penumbra and trigger brain inflammation and oxidative stress within days; (iii) The inflammatory condition and reactive oxygen species trigger necrosis and apoptosis of brain cells through mitochondrial and DNA damage for days and weeks. In addition, disruption of the blood-brain barrier (BBB) is one of the secondary events in brain injury and progression of cerebral ischemia [71,72]. BBB disruption is a major pathophysiological contributor to brain injury through the evolution of cerebral ischemia and is regulated by inflammatory modulators, oxidative damage, and altered regulation of adhesion molecules [17,73,74].

Additional Reference

  1. Dirnagl, U.; Iadecola, C.; Moskowitz, M.A. Pathobiology of ischaemic stroke: an integrated view. Trends Neurosci 1999, 22, 391-397, doi:10.1016/s0166-2236(99)01401-0.
  2. Pedata, F.; Dettori, I.; Coppi, E.; Melani, A.; Fusco, I.; Corradetti, R.; Pugliese, A.M. Purinergic signalling in brain ischemia. Neuropharmacology 2016, 104, 105-130, doi:10.1016/j.neuropharm.2015.11.007.
  3. Li, Y.; Zhong, W.; Jiang, Z.; Tang, X. New progress in the approaches for blood-brain barrier protection in acute ischemic stroke. Brain Res Bull 2019, 144, 46-57, doi:10.1016/j.brainresbull.2018.11.006.
  4. Sifat, A.E.; Vaidya, B.; Abbruscato, T.J. Blood-Brain Barrier Protection as a Therapeutic Strategy for Acute Ischemic Stroke. The AAPS Journal 2017, 19, 957-972, doi:10.1208/s12248-017-0091-7.
  5. Kassner, A.; Merali, Z. Assessment of Blood-Brain Barrier Disruption in Stroke. Stroke 2015, 46, 3310-3315, doi:10.1161/STROKEAHA.115.008861.

COMMENT 2: For the introduction, the author should emphasize the importance and significance of the intervention and treatment of stroke from the perspective of metabolism.

RESPONSE: As per your comments, we added the information for the treatment of stroke from the perspective of metabolism in page 3 as follows:

From the perspective of therapeutic considerations, preclinical studies addressing the neuroprotective potential of manipulating certain metabolites associated with various pathways of cerebral ischemia include excitotoxic metabolites using a peptide inhibitor of c-Jun [33], anaerobic glycolysis-induced lactic acidosis using dichloroacetate [34] or induction of normoglycemia [35], and proinflammatory pathway mediators (lysophosphatidylcholines (lysoPCs) and acylcarnitines) using a synthetic agonist for RXR-Nurr1 heterodimer complex [36]. The approach to treat cerebral ischemia from the metabolism perspective is an important methodological consideration that differs from the current treatment methods that are based on mitigating the etiology of stroke by preventing or opening up the vascular occlusion. Unfortunately, only few studies have reported on the treatment of cerebral ischemia from the metabolism perspective, due to incomplete understanding of the metabolic changes occurring in the ischemic brain.

Additional References

  1. Borsello, T.; Clarke, P.G.; Hirt, L.; Vercelli, A.; Repici, M.; Schorderet, D.F.; Bogousslavsky, J.; Bonny, C. A peptide inhibitor of c-Jun N-terminal kinase protects against excitotoxicity and cerebral ischemia. Nat Med 2003, 9, 1180-1186, doi:10.1038/nm911.
  2. Kaplan, J.; Dimlich, R.V.; Biros, M.H. Dichloroacetate treatment of ischemic cerebral lactic acidosis in the fed rat. Ann Emerg Med 1987, 16, 298-304, doi:10.1016/s0196-0644(87)80175-0.
  3. Wagner, K.R.; Kleinholz, M.; de Courten-Myers, G.M.; Myers, R.E. Hyperglycemic versus normoglycemic stroke: topography of brain metabolites, intracellular pH, and infarct size. J Cereb Blood Flow Metab 1992, 12, 213-222, doi:10.1038/jcbfm.1992.31.
  4. Loppi, S.; Kolosowska, N.; Karkkainen, O.; Korhonen, P.; Huuskonen, M.; Grubman, A.; Dhungana, H.; Wojciechowski, S.; Pomeshchik, Y.; Giordano, M., et al. HX600, a synthetic agonist for RXR-Nurr1 heterodimer complex, prevents ischemia-induced neuronal damage. Brain Behav Immun 2018, 73, 670-681, doi:10.1016/j.bbi.2018.07.021.

COMMENT 3: It will be very helpful for the author to briefly introduce the change of the major glucose metabolic pathways including glycolysis, TCA cycle, Pentose-phosphate pathway before and after ischemic stroke, which lead to the metabolites change.

RESPONSE: As per your recommendation, we briefly added about the changes of glycolysis, TCA cycle, Pentose-phosphate pathways in cerebral ischemia condition in page 4 as follows:

Alterations in metabolite levels have been reported, including organic acids, amino acids, free fatty acids, lipids, and low-density lipoprotein in the serum and plasma of cerebral ischemia patients. Basically, glucose metabolic pathways are highly affected by cerebral ischemia due to reduction in oxygen and nutrient availability [66]. Characteristically, glycolysis is changed from aerobic to anaerobic pathway in cerebral ischemia [67]. In addition, the tricarboxylic acid (TCA) cycle is suppressed by oxidative radicals due to the predominance of anaerobic glycolysis [68]. Moreover, the pentose-phosphate pathway is activated as an endogenous antioxidant mechanism by increasing nicotinamide adenine dinucleotide phosphate (NADPH) / nicotinamide adenine dinucleotide (NAD)+ ratio [69].

Additional References

  1. Rink, C.; Khanna, S. Significance of brain tissue oxygenation and the arachidonic acid cascade in stroke. Antioxid Redox Signal 2011, 14, 1889-1903, doi:10.1089/ars.2010.3474.
  2. Schurr, A. Lactate, glucose and energy metabolism in the ischemic brain (Review). Int J Mol Med 2002, 10, 131-136, doi:10.3892/ijmm.10.2.131.
  3. Sahni, P.V.; Zhang, J.; Sosunov, S.; Galkin, A.; Niatsetskaya, Z.; Starkov, A.; Brookes, P.S.; Ten, V.S. Krebs cycle metabolites and preferential succinate oxidation following neonatal hypoxic-ischemic brain injury in mice. Pediatr Res 2018, 83, 491-497, doi:10.1038/pr.2017.277.
  4. Imahori, T.; Hosoda, K.; Nakai, T.; Yamamoto, Y.; Irino, Y.; Shinohara, M.; Sato, N.; Sasayama, T.; Tanaka, K.; Nagashima, H., et al. Combined metabolic and transcriptional profiling identifies pentose phosphate pathway activation by HSP27 phosphorylation during cerebral ischemia. Neuroscience 2017, 349, 1-16, doi:10.1016/j.neuroscience.2017.02.036.

COMMENT 4: It will be also very helpful for the author to introduce the change of the metabolites in blood after cerebral ischemia.

RESPONSE: In line with your comment, we summarized about the analyzed metabolites in blood in page 4 and 6 as follows:

Page 4

Alterations in metabolite levels have been reported, including organic acids, amino acids, free fatty acids, lipids, and low-density lipoprotein in the serum and plasma of cerebral ischemia patients.

Page 6

Changes in metabolite levels have been reported, including amino acids, free fatty acids, lipids, nucleotides, short peptides, phosphoethanolamine (PE), phosphoserine (PS), and phosphocholines (PCs) in the plasma, brain tissue, and cerebrospinal fluid in MCAo rodent models.

COMMENT 5: Glutamate/Glutamine coupling between astrocyte and neurons are very important for brain metabolism homeostasis. Accumulating evidence suggested that cerebral ischemia leads to neuronal and glial metabolic uncoupling, in which there is an acute increase in extracellular glutamate levels. It will be helpful for the author to include this information in this review article.

RESPONSE: We added the information of the Glutamate/Glutamine coupling between astrocyte and neurons in page 5 as follows:

In particular, the increase in the excitotoxic neurotransmitter glutamate disrupts the balance of glutamate/glutamine coupling, which is important for brain metabolic homeostasis between astrocytes and neurons [81].

Additional Reference

  1. Shen, J. Modeling the glutamate-glutamine neurotransmitter cycle. Front Neuroenergetics 2013, 5, 1, doi:10.3389/fnene.2013.00001.

COMMENT 6: The Microdialysis coupled with LC/MS or HPLC are the classical investigation methods for the analysis of extracellular metabolites after a variety of brain injuries, including cerebral ischemia. The author should also introduce this method.

RESPONSE: As per your comment, we added the introduction of microdialysis in page 3 as follows:

In the case of cerebral ischemia, this can be done using microdialysis, nuclear magnetic resonance (NMR), and MS [13,14,40,41]. Microdialysis is used for continuous measurement of molecules, including neurotransmitters, hormones, and metabolites in the extracellular fluid coupled with liquid chromatography [42,43].

Additional References

  1. Schulz, M.K.; Wang, L.P.; Tange, M.; Bjerre, P. Cerebral microdialysis monitoring: determination of normal and ischemic cerebral metabolisms in patients with aneurysmal subarachnoid hemorrhage. J Neurosurg 2000, 93, 808-814, doi:10.3171/jns.2000.93.5.0808.
  2. Chefer, V.I.; Thompson, A.C.; Zapata, A.; Shippenberg, T.S. Overview of brain microdialysis. Curr Protoc Neurosci 2009, Chapter 7, Unit7 1, doi:10.1002/0471142301.ns0701s47.
  3. Liu, K.; Lin, Y.; Yu, P.; Mao, L. Dynamic regional changes of extracellular ascorbic acid during global cerebral ischemia: studied with in vivo microdialysis coupled with on-line electrochemical detection. Brain Res 2009, 1253, 161-168, doi:10.1016/j.brainres.2008.11.096.

COMMENT 7: For the “Summary of metabolites associated with MCAo in the rat model”, maybe some information regarding the metabolite change in the rat stroke model is still missing. For example, some important metabolites such as N-acetyl-Aspartate, NAD+, NADH, NADP+, NADPH, etc, the most commonly reported metabolites in stroke studies, had not been introduced in the present review article.

RESPONSE: We added the suggested points in Table 2 and page 6 as follows:

Additionally, changes in the levels of N-acetyl-aspartate (NAA) serve as an important biomarker for cerebral ischemia [115]. NAA is a source for acetyl groups and is related to neurotransmitter metabolism [116]. Moreover, NAA is a marker of neuronal integrity because it is synthesized in the neuronal mitochondrial membrane [117]. In addition, NADP+/NADPH and NAD+/NADH represent redox status, energy dependent process, and energy phosphate stores [118]. Future metabolomic studies should be focused on these important metabolites.

Additional References

  1. Li, M.-H.; Ruan, L.-Y.; Chen, C.; Xing, Y.-X.; Hong, W.; Du, R.-H.; Wang, J.-S. Protective effects of Polygonum multiflorum on ischemic stroke rat model analysed by 1H NMR metabolic profiling. Journal of Pharmaceutical and Biomedical Analysis 2018, 155, 91-103, doi:10.1016/j.jpba.2018.03.049.
  2. Birken, D.L.; Oldendorf, W.H. N-Acetyl-L-Aspartic acid: A literature review of a compound prominent in 1H-NMR spectroscopic studies of brain. Neuroscience & Biobehavioral Reviews 1989, 13, 23-31, doi:10.1016/S0149-7634(89)80048-X.
  3. Baslow, M.H. N-acetylaspartate in the vertebrate brain: metabolism and function. Neurochem Res 2003, 28, 941-953, doi:10.1023/a:1023250721185.
  4. Esumi, K.; Nishida, M.; Shaw, D.; Smith, T.W.; Marsh, J.D. NADH measurements in adult rat myocytes during simulated ischemia. Am J Physiol 1991, 260, H1743-1752, doi:10.1152/ajpheart.1991.260.6.H1743

Reviewer 2 Report

This is an interesting manuscript reviewing the metabolic and cellular metabolites/markers of cerebral ischemia in patients and rats.
Major concerns:
-the authors should change a title because the manuscript discusses mostly methods and markers but not the mechanisms.
-the authors should clarify that only rats' animal models were discussed and explain why other rodents (specifically transgenic mice) data were not included in the review.
Minor:
-Fig 1B, legend:
-“suture” should be changed to “filament”;
-“TTC-stained frontal (upper left panel) to posterior (bottom right panel) brain sections” should be explained as “excluding cerebellum” or edited as “cerebral sections”;
-“Scale bar= 5 mm” should be corrected.
Lines 313-315: “In addition, animal models of cerebral ischemia can be complicated and are often time-consuming, especially with the need for validation of ischemic pathophysiological condition, compared to in vitro models.” and lines 354-356 “In addition, the maintenance of an animal model is time-consuming compared to in vitro models, which are suitable for high-throughput low-cost analyses and can have promising possibilities for development.” should be edited because the maintenance of the in vitro experiments with cultured cells as well as maintenance of the cell cultures and conditions are also difficult and time-consuming.

Author Response

Answers to the comments

Reviewer # 2

This is an interesting manuscript reviewing the metabolic and cellular metabolites/markers of cerebral ischemia in patients and rats.

Major concerns:

COMMENT 1: the authors should change a title because the manuscript discusses mostly methods and markers but not the mechanisms.

RESPONSE: In line with your comment, we changed the title as follows:

Metabolome Changes in Cerebral Ischemia

COMMENT 2: the authors should clarify that only rats' animal models were discussed and explain why other rodents (specifically transgenic mice) data were not included in the review.

RESPONSE: As per your suggestion, we added the introduction for other rodent models. It is to be noted that there are insufficient studies available on the metabolic profile of mouse cerebral ischemia models nor transgenic mice. We have added these points in page 6 as follows:

Small animals can be challenging to handle and model for cerebral ischemia, hence larger animals are often used for this purpose [102]. In addition, infarct size is known to be affected by rodent strain [103]. For example, spontaneously hypertensive rats exhibit large infarcts with low variability, whereas Sprague-Dawley rats have relatively small infarcts with high variability with MCAo [104]. As expected, infarct volume correlates with mortality rate [105], and high variability reduces statistical power [106]. In the case of mouse strains, C57B1/6 mice are reported to generate relatively larger infarcts compared with Sv129 mice [107]. Animal models of cerebral ischemia include the intraluminal filament MCAo, direct MCA ligation by a subtemporal craniectomy approach, photothrombosis, endothelin-1, and the embolic stroke model [103,108-110]. Among these, the rodent intraluminal filament MCAo model is the most commonly used because of the high similarity with human cerebral ischemia, exhibits a penumbra, is highly reproducible, has a highly controllable reperfusion, and does not require craniectomy [103]. However, the MCAo model also has limitations such as hyper-/hypothermia issues, increased hemorrhage with certain filament types, and the fact that it is unsuitable for thrombolysis studies. In addition, transgenic and knockout mice have been developed and used for studies of genomic and proteomic mechanisms of cell injury, neuroprotection, and redox mechanisms of cerebral ischemia [111,112].

Additional References

  1. Sommer, C.J. Ischemic stroke: experimental models and reality. Acta Neuropathol 2017, 133, 245-261, doi:10.1007/s00401-017-1667-0.
  2. Fluri, F.; Schuhmann, M.K.; Kleinschnitz, C. Animal models of ischemic stroke and their application in clinical research. Drug Des Devel Ther 2015, 9, 3445-3454, doi:10.2147/DDDT.S56071.
  3. Walberer, M.; Stolz, E.; Muller, C.; Friedrich, C.; Rottger, C.; Blaes, F.; Kaps, M.; Fisher, M.; Bachmann, G.; Gerriets, T. Experimental stroke: ischaemic lesion volume and oedema formation differ among rat strains (a comparison between Wistar and Sprague-Dawley rats using MRI). Lab Anim 2006, 40, 1-8, doi:10.1258/002367706775404426.
  4. Murata, Y.; Wang, X.; Lo, E. A Thromboembolic Rat Model of Focal Cerebral Ischemia and Reperfusion with tPA. 2009; 10.1007/978-1-60327-185-1_13pp. 155-167.
  5. Strom, J.O.; Ingberg, E.; Theodorsson, A.; Theodorsson, E. Method parameters' impact on mortality and variability in rat stroke experiments: a meta-analysis. BMC Neurosci 2013, 14, 41, doi:10.1186/1471-2202-14-41, doi:10.1186/1471-2202-14-41.
  6. Maeda, K.; Hata, R.; Hossmann, K.A. Regional metabolic disturbances and cerebrovascular anatomy after permanent middle cerebral artery occlusion in C57black/6 and SV129 mice. Neurobiol Dis 1999, 6, 101-108, doi:10.1006/nbdi.1998.0235.
  7. Liu, F.; McCullough, L.D. Middle cerebral artery occlusion model in rodents: methods and potential pitfalls. J Biomed Biotechnol 2011, 2011, 464701, doi:10.1155/2011/464701.
  8. Traystman, R.J. Animal models of focal and global cerebral ischemia. ILAR J 2003, 44, 85-95, doi:10.1093/ilar.44.2.85.

Minor:

Fig 1B, legend:

COMMENT 3: “suture” should be changed to “filament”;

RESPONSE: As you pointed out, we changed “suture” to “filament.”

COMMENT 4: “TTC-stained frontal (upper left panel) to posterior (bottom right panel) brain sections” should be explained as “excluding cerebellum” or edited as “cerebral sections”;

RESPONSE: As you suggested, this is now edited to specify “cerebral cortex” on page 14.

COMMENT 5: “Scale bar= 5 mm” should be corrected.

RESPONSE: We have deleted the scale bar.

COMMENT 6: Lines 313-315: “In addition, animal models of cerebral ischemia can be complicated and are often time-consuming, especially with the need for validation of ischemic pathophysiological condition, compared to in vitro models.” and lines 354-356 “In addition, the maintenance of an animal model is time-consuming compared to in vitro models, which are suitable for high-throughput low-cost analyses and can have promising possibilities for development.” should be edited because the maintenance of the in vitro experiments with cultured cells as well as maintenance of the cell cultures and conditions are also difficult and time-consuming.

RESPONSE: We changed the mentions about comparisons between in vivo and in vitro in page 11 and 20 as follows:

Page 11

In addition, animal models of cerebral ischemia can be complicated and are often time-consuming, especially with the need for validation of ischemic pathophysiological condition. Therefore, the development of in vitro based methods is useful for initial rapid evaluation and as supplementary to experiments in animal models.

Page 20

However, there are some limitations in animal models, such as genomic differences between humans and rodents as well as differences in biological phenotypes, plus the time and labor needed to generate and maintain animal models

Round 2

Reviewer 2 Report

Minor:
Fig 1B, legend:
"Schematic diagram of the surgical procedure of MCAo model, and coronal sections (2 mm thick) of cerebral cortex of MCAo lesioned rat brains ..."
should be edit as
"Schematic diagram of the surgical procedure of MCAo model, and cerebral coronal sections (2 mm thick) of MCAo lesioned rat brains ...."

Author Response

Reviewer # 2

COMMENT: Fig 1B, legend:"Schematic diagram of the surgical procedure of MCAo model, and coronal sections (2 mm thick) of cerebral cortex of MCAo lesioned rat brains ..."

should be edit as

"Schematic diagram of the surgical procedure of MCAo model, and cerebral coronal sections (2 mm thick) of MCAo lesioned rat brains ...."

RESPONSE: As per your comments, we changed Figure 1B legend in page 14.
